# Crosstalk between CXCR4/ACKR3 and EGFR Signaling in Breast Cancer Cells

**DOI:** 10.3390/ijms231911887

**Published:** 2022-10-06

**Authors:** Maria Neves, Viviana Marolda, Federico Mayor, Petronila Penela

**Affiliations:** 1Department of Molecular Biology, University Institute of Molecular Biology (IUBM-UAM), Centre for Molecular Biology “Severo Ochoa” (CBMSO) UAM-CSIC, Autonomous University of Madrid and Health Research Institute Hospital Universitario La Princesa, 28006 Madrid, Spain; 2Health Research Institute Hospital Universitario La Princesa, 28006 Madrid, Spain; 3CIBER of Cardiovascular Diseases (CIBERCV), Institute of Health Carlos III (ISCIII), 28006 Madrid, Spain

**Keywords:** breast cancer, chemokines, CXCR4, ACKR3, CXCL12, GRK2, EGF receptors, MAPK signal crosstalk, Her2

## Abstract

A better understanding of the complex crosstalk among key receptors and signaling pathways involved in cancer progression is needed to improve current therapies. We have investigated in cell models representative of the major subtypes of breast cancer (BC) the interplay between the chemokine CXCL12/CXCR4/ACKR3 and EGF receptor (EGFR) family signaling cascades. These cell lines display a high heterogeneity in expression profiles of CXCR4/ACKR3 chemokine receptors, with a predominant intracellular localization and different proportions of cell surface CXCR4+, ACKR3+ or double-positive cell subpopulations, and display an overall modest activation of oncogenic pathways in response to exogenous CXCL12 alone. Interestingly, we find that in MDA-MB-361 (luminal B subtype, Her2-overexpressing), but not in MCF7 (luminal A) or MDA-MB-231 (triple negative) cells, CXCR4/ACKR3 and EGFR receptor families share signaling components and crosstalk mechanisms to concurrently promote ERK1/2 activation, with a key involvement of the G protein-coupled receptor kinase 2 (GRK2) signaling hub and the cytosolic tyrosine kinase Src. Our findings suggest that in certain BC subtypes, a relevant cooperation between CXCR4/ACKR3 and growth factor receptors takes place to integrate concurrent signals emanating from the tumor microenvironment and foster cancer progression.

## 1. Introduction

Breast cancer (BC) is the most prevalent cancer among women worldwide [1]. BC is a heterogeneous disease characterized by multiple molecular signatures, genetic and genomic variations [2]. BC is categorized the following established criteria: tumor morphology, grade, stage, and expression of key genes, such as estrogen receptor-α (ER), progesterone receptor (PR), and human epidermal growth factor receptor-2 (HER2 or ERBB2), to determine adequate treatment options. Based on their distinctive molecular profiling, breast cancers can be grouped in different subtypes: estrogen-receptor positive luminal, ERBB2+ (or HER2+), basal-like (also referred as triple negative due to absence of ER, PR and HER2) and claudin-low [3,4,5,6]. Further subclassification of luminal tumors using a Ki-67 index, HER2 status, and PR-receptor status identifies low- and high-risk categories corresponding to luminal A (HER2 negative, Ki-67 low, and PR high) and luminal B (HER2 negative, Ki-67 high, PR-negative) or luminal B-like (HER2+, any PR level, any Ki-67 level). Regardless of treatment options available, resistance to therapy is common among all subtypes, either due additional mutations or because of the rewiring of the complex interactions/crosstalk among key signaling cascades in BC progression [7,8]. These limitations in current therapies highlight the need for deeper understanding of complex pathways leading to BC development and progression.

The chemokine receptor system plays a relevant role in BC, where some chemokines and their cognate receptors, namely CXCL12 and its receptors CXCR4 and ACKR3 are found overexpressed and often dysregulated in both tumor and tumor microenvironment (TME) cells [9,10,11]. The CXCL12/CXCR4/ACKR3 signaling axis has been implicated in tumor cell growth, survival, invasion and metastasis, immune evasion, and angiogenesis in BC and other tumor cell types, thus leading to increased proliferation and fostering aggressiveness by promoting neovascularization and metastatic spread [10,11]. Emerging data obtained from distinct cancer cell types with endogenous chemokine receptor levels highlights a very complex, heterogeneous scenario, where multiple variables must be considered. Context-specific modulation of ligand level, receptor patterns of expression and subcellular location, interactomes specific of each tumor type and crosstalk mechanisms with other receptors and components of TME dictate receptor responses [11,12,13,14,15,16]. In fact, a frequent characteristic of tumor cells is the concomitant overexpression of CXCR4/ACKR3 and growth factor receptors (GFR), an important element in cell proliferation and survival. Several GFR are involved in BC progression such as the ErbB receptor family [17]. Alterations in the ErbB family (namely ErbB1 (EGFR/HER1), the orphan receptor ErbB2 (HER2 or Neu), ErbB3 (HER3), and ErbB4 (HER4)) have been associated with BC incidence and poor prognosis [18,19] and mutations of ErbB effectors are among the most common genetic abnormalities associated with breast cancer [20]. ErbB receptors signal through MAPK, AKT, and other pathways ultimately activating transcription factors that promote expression of genes encoding growth factors, cyclins, and cytokines, dictating cell proliferation, differentiation, apoptosis, and motility.

Given the frequent concurrent up-regulation of CXCR4/ACKR2 and ErbB family receptors in BC and their shared relevance for tumor growth and metastasis occurrence, investigating their interplay mechanisms in tumor contexts is of utmost importance [11].

In this regard, the G protein-coupled receptor kinase 2 (GRK2) can act as regulator of both G protein-coupled receptors (GPCRs), including CXCR4/ACKR3, and GFR of the receptor tyrosine kinase (RTKs) family, and has emerged as a critical signaling hub involved in orchestrating most tumor hallmarks [21,22,23]. Deregulation of GRK2 levels and activity are reported in pathological contexts such as metabolic dysfunctions and several types of cancer, namely breast [22,24]. Indeed, increased expression of GRK2 potentiate several malignant features of breast cancer cells, and its levels positively correlate with tumor growth and increased metastasis occurrence [25]. Even though a positive role of GRK2 in ErbB-RTKs signaling has been described, the impact of GRK2 in the modulation of GPCRs such as CXCR4 and ACKR3 and in their interplay with GF-RTKs have yet to be addressed. We have recently shown that GRK2 and EGFR can modulate CXCR4-mediated Gi activation in HEK-293 cells by mechanisms involving tyrosine phosphorylation, in line with other reports on GFR-GPCR crosstalk [26,27,28]. Hence, we have further investigated the complexity of CXCL12/CXCR4/ACKR3 and EGF-RTKs signaling interplay and explored the role of GRK2 in such crosstalk in different BC cell lines corresponding either to tumors amenable to targeted therapy with low (luminal A) or high (luminal B) risk of relapse or difficult-to-treat tumours with a poor prognosis (triple-negative).

## 2. Results

### 2.1. Breast Cancer Cell Lines Display Heterogeneous Patterns of CXCR4/ACKR3 Distribution

Enhanced CXCR4 and ACKR3 levels in breast cancer compared to normal tissues has been reported, although their relative abundance within molecular subtypes and in BC cellular models is unclear [29]. We thus used flow cytometry to characterize their total and plasma membrane-bound protein levels in well-established model cell lines representative of the main BC types relative to treatment options and responsiveness: MDA-MB-361 (Luminal B, Her2+ subtype, targeted therapy, high risk), MCF7 (luminal A subtype, targeted therapy, low risk) and MDA-MB-231 (triple negative subtype, chemotherapy, high risk). All BC cell lines subtypes expressed CXCR4 and ACKR3 although with different levels available for extracellular ligand binding, with an important internalized receptor pool (Figure 1). The MDA-MB-361 cell line expresses slightly more CXCR4 in the plasma membrane, while MDA-MB-231 is the one with lowest surface receptors present (Figure 1A). Total protein CXCR4 levels showed no major differences among subtypes, in line with mRNA expression data (Appendix A). As for ACKR3 (Figure 1B), the membrane-bound fraction is very low in most cell lines, especially in MDA-MB-231, whereas total ACKR3 appears to be more variable at both protein and mRNA levels (Appendix A). The very high proportion of internalized CXCR4 and ACKR3 receptors in BC cell lines might be a consequence of agonist-induced endocytosis due to autocrine production and secretion of CXCL12. The quantification of CXCL12 in cell-conditioned media (Appendix A) indicates that all BC lines secrete different levels of this chemokine, although these do not overall correlate with the extent of internalized receptors, thereby suggesting that other signaling components in BC cancer cells might contribute to shape a steady-state pool of intracellular receptors.

Our results also show that whereas all cell lines display populations where both receptors are found co-localized, the proportion of these populations vary greatly among different BC cell types (Figure 2). Moreover, the presence of cells with given single receptors differ from cell type to type, thus resulting in a very heterogeneous pattern of receptor distribution. Considering total levels (i.e., receptor at surface plus internalized) we detected in MCF7 and MDA-MB-231 cells a higher proportion of cells displaying both receptors, followed by a subset that expresses only ACKR3. MDA-MB-361 cells displayed an opposite subpopulation distribution of total receptors, with the majority of cells being only positive for ACKR3. The most striking differences appeared when receptors that reside at the plasma membrane in steady-state conditions were quantified. Firstly, for each one BC cell type more than 50% of the cells have no detectable CXCL12 receptors at their surface, likely reflecting constitutive receptor internalization and/or altered receptor trafficking processes. Secondly, a relevant subset of surface-CXCR4+ cells is only present in MDA-MB-361 cells. However, plasma membrane ACKR3+ and double ACKR3 + CXCR4+ subpopulations were detected in all BC cell types.

Overall, our results put forward a very variable presence of surface CXCR4 and ACKR3 across different BC cell lines and complex patterns of CXCR4/ACKR3 distribution and co-localization within cell lines, suggesting that the response of BC cells to CXCR4/ACKR3 ligands might be very heterogenous.

### 2.2. Breast Cancer Cells Differentially Activate ERK in Response to CXCL12 and ErbB Receptor Ligands

Different downstream pathways, such as the mitogen-activated protein kinases (MAPK) cascades, convey CXCL12 signal via CXCR4/ACKR3 with roles in tumor biology [30,31,32]. Hence, we investigated whether the heterogeneous distribution of endogenous CXCR4/ACKR3 receptors in our BC cell lines may correlate with ERK1/2 activation patterns in response to exogenous CXCL12. Analysis of ERK1/2 phosphorylation status indicated a higher level of basal ERK1/2 stimulation in MDA-MB-231 cells compared to the other cell types (Figure 3A), consistent with the reported presence of activating mutations of Ras and Raf in this line [33]. Surprisingly, global ERK1/2 stimulation by exogenous CXCL12 was very low across the BC cell lines (Figure 3B). MDA-MB-361 was the only cell model that displayed a transient significative activation of ERK1/2 (maximal stimulation observed at 2 min). This fact might be related to the different subpopulation patterns of surface CXCL12 receptors in this BC cell line (with higher proportion of CXCR4+-only cells) and/or to the lower levels of basal ERK1/2 activation status, thus allowing a more dynamic activation range [33].

Recent reports describe bidirectional cross-activation mechanisms between CXCR4/ACKR3 and RTKs that result in ligand-independent constitutive proliferation and survival loops [11,14,16]. This scenario would also contribute to explain the low responsiveness of BC cells to exogenous CXCL12. Thus, we explored whether CXCR4/ACKR3 and ERBB receptor family interplay would impact on ERK1/2 activation. We found that simultaneous stimulation of MDA-MB-361 cells with EGF and CXCL12 promoted a higher ERK1/2 activating phosphorylation pattern, with a significant increase in the area under the curve (AUC) compared to EGF or CXCL12 alone (Figure 3C). Interestingly, a similar trend was observed upon co-stimulation with heregulin (Appendix A) a specific ligand for the HER3 receptor and HER2 co-receptor which are both highly expressed in MDA-MB-361 cells (Appendix A).

Conversely, in MCF7 and MDA-MB-231 cells, no potentiation of ERK1/2 phosphorylation was observed when EGF and CXCL12 were combined (Appendix A), but rather a faster decay (MCF7) or attenuation (MDA-MB-231) of the activation pattern of this pathway. This suggests that signaling amplification effects between these receptors not only require high levels of EGF but a cell-type specific repertoire of EGFR and HER2·receptors and downstream transducers and regulators. We thus decided to focus our research on CXCR4/ACKR3 signaling and crosstalk mechanisms in the MDA-MB-361 BC cell model, a subtype characterized by lower basal levels of active ERK and high levels of EGFR and HER2 heterodimers, and which displayed a positive crosstalk of ErbB-RTKs and CXCR4/ACKR3 to the ERK pathway.

### 2.3. Exogenous CXCL12 Fails to Activate Other Key Transducing Molecules Related to Oncogenic Pathways in MDA-MB-361 Cells

We investigated additional effectors previously reported as being activated by CXCL12/CXCR4/ACKR3 in other experimental settings, such as AKT [34,35], Src [34,36,37], S6 [38], focal adhesion kinase (FAK) [39,40] and p38 MAPK [41,42]. Their activation was assessed in MDA-MB-361 cells at 2 min after CXCL12 stimulation (Figure 4A) which corresponds with the peak activation of ERK. None of these effectors were readily activated over steady-state levels by exogenous CXCL12 stimulation. To rule out that activation of these effectors had a different time window of maximal response to CXCL12 compared to ERK1/2, extended time-courses were analyzed without detecting changes. To test whether CXCL12 fails to activate the above-mentioned effectors due to ligand-independent activation of downstream transducing components, cells were treated with EGF. This factor promoted a significant and rapid activation of ERK1/2, AKT, Src, S6 and p38 pathways in MDA-MB-361 cells (Figure 4B), suggesting that pathways engaging these effectors are within the dynamic range of activation and that external stimuli other than CXCL12 can overcome the basal thresholds.

We hypothesized that the CXCR4/ACKR3 axis could be in a constitutively active status via autocrine loops, and therefore addition of exogenous CXCL12 could not further activate the receptors. To test this, MDA-MB-361 cells were treated with ACKR3 (nanobody VUN700) or CXCR4 inhibitors (nanobody VUN400 or AMD3100) (Appendix A). Inhibition of CXCR4 decreased steady state activation of AKT and SRC pathways, and ACKR3 inhibition showed a similar trend. These data point to the fact that CXCR4 and ACKR3 contribute to a certain extent to the activation of AKT and SRC cascades in the absence of an exogenous ligand [15].

### 2.4. Mechanisms Involved in the Crosstalk between the CXCL12/CXCR4/ACKR3 and EGFR Family Signaling in Breast Cancer Cells Lines

The peak of ERK1/2 activation induced by co-stimulated CXCR4/ACKR3 and EGFR was detected at 2 min in MDA-MB-361 cells. As for either heregulin (HRG) or HRG combined with CXCL12, maximal responses were observed at 5 min, although the intensity of ERK1/2 activation was already significantly amplified at earlier time points when compared to peak-activated ERK1/2 by either ligand alone. For the sake of simplicity, we confirmed with all ligands and combinations that these potentiating effects are observed at 2 min of stimulation (Figure 5A), and choose this time point to further investigate the molecular mechanisms potentially underlying EGFR family crosstalk with CXCR4/ACKR3.

Since GRK2 is a positive modulator of GF-RTK signaling [26] in addition to its role in GPCR regulation [16,21,43], we assessed its role in CXCR4/ACKR3 and EGFR family signaling and crosstalk in MDA-MB-361 cells by using different approaches involving pharmacological treatments with CMPD101, an inhibitor of GRK2/3 kinase activity, and heterologous expression of extra levels of wild-type GRK2. The low CXCL12-induced ERK1/2 activation was not significantly altered in the presence of CMPD101 (Figure 5B). On the contrary, the robust EGF- or HRG-dependent ERK1/2 activation were significantly decreased by CMPD101, confirming that GRK2 plays a positive role in GF-RTK signaling as previously reported in other BC cell lines [25]. Similarly, GRK2 inhibition prior to concomitant co-stimulation with GF ligands and CXCL12 decreases completely (HRG) or partially (EGF) the extent of ERK1/2 activation (Figure 5B). Interestingly, extra levels of GRK2 protein tend to down-modulate ERK1/2 stimulation by CXCL12 alone (Appendix A), consistent with its previously reported role in CXCR4/ACKR3 desensitization [26]. The fact that pharmacological inhibition of GRK2 has no apparent effect on CXCL12 signaling (Figure 5B) may suggest a catalytic-independent desensitization mechanism. GRK2 overexpression enhanced ERK1/2 stimulation by EGF alone, and co-stimulation with EGF and CXCL12 in GRK2-overexpressing cells led to a significantly stronger activation of ERK1/2 compared to each stimulus alone (Appendix A). Taken together, these results support an overall positive role for GRK2 in ERK1/2 activation when both CXCL12 and EGFR are co-stimulated.

Next, we sought to characterize deeper the mechanism of action of the CXCR4/ACKR3 and EGF-RTK crosstalk, by pharmacological inhibition of known downstream signaling components associated with each individual receptor. As expected, inhibition of CXCR4 with AMD3100, a small-molecule antagonist, or VUN400, a nanobody antagonist and inhibitor of CXCL12 receptor binding [44], resulted in decreased ERK activation by CXCL12. Surprisingly, both CXCR4 inhibitors also decreased EGF-mediated ERK activation and consequently, the additive effect of CXCL12 and EGF co-treatment on MAPK kinase stimulation (Figure 6). The effect of CXCR4 inhibition in HRG signaling to ERK1/2 was less pronounced, but a similar tendency was noted. Akin to CXCR4 inhibitors but less strongly, the ACKR3 nanobody inhibitor VUN700 decreased CXCL12, EGF, HRG or combined ERK, suggesting that GF-RTK activation could also involve ACKR3-mediated pathways (Figure 6).

Of note, treatment with pertussis toxin (PTX), a toxin targeting Gi/o proteins and preventing their interaction with GPCR, only slightly decreased CXCL12-mediated ERK activation, consistent with the notion that CXCR4 or ACKR3 also couple to G protein-independent routes for biased MAPK activation. Strikingly, PTX treatment decreased EGF or HRG-mediated ERK activation and disrupted their crosstalk with CXCL12-coupled receptors (Figure 6). Taken together with previous observations, these results suggest that EGF-RTKs signal partially to ERK through CXCR4 in a G-protein dependent manner.

On the GF-RTK side, treatment with AG1478, a small-molecule inhibitor of EGFR tyrosine kinase activity, significantly decreased ERK activation by EGF or HRG (to levels close to unstimulated cells) in all conditions tested. Of note, AG1478 slightly decreased CXCL12-mediated ERK activation, indicating that a component of CXCR4/ACKR3 signaling depends on EGF-RTK (Figure 6). Overall, these data suggested that EGF-RTK signaling to ERK, either by direct stimulation or via crosstalk with CXCR4/ACKR3, is entirely dependent on their kinase activity.

Src is a proto-oncogen that triggers transformation and tumor progression [45], and a downstream mediator of both GF-RTK and GPCR signaling. Treatment with the specific Src inhibitor PP2 decreased CXCL12, EGF or HRG-mediated ERK activation, implying that this kinase is a common downstream effector engaged by CXCR4/ACKR3, EGFR and HER2/3 in the MAPK pathway. Remarkably, potentiation of ERK phosphorylation by co-stimulation of CXCR4/ACKR3-EGFR or CXCR4/ACKR3-HER2/3 was abolished in the presence of PP2 (Figure 6), suggesting that in MDA-MB-361 cells cytosolic tyrosine kinases are also involved in their crosstalk. Taken together, our results indicate that CXCR4/ACKR3 and EGF-RTKs display both alternative and common pathways of MAPK activation which are amplified when receptors are co-stimulated.

## 3. Discussion

We find that growth factor receptors and CXCR4/ACKR3 chemokine receptors cooperate to integrate signals from the tumor microenvironment in certain breast cancer subtypes. The CXCL12/CXCR4/ACKR3 cascade is reported to be involved in almost every aspect of breast cancer tumorigenesis [11,14,46]. However, the study of the underlying molecular mechanisms is very challenging in endogenous cancer contexts. Canonical signaling pathways downstream of these GPCRs have been mostly described in immune cells or heterologous model systems, such as HEK293 cells. These experimental settings are helpful to understand specifics of receptor activation and possible molecular interactions and modulation, but lack the context often found in oncogenic environments, such as tumor-specific interactomes, varying levels of endogenous receptors and intricate crosstalk mechanisms with other transduction networks taking place in the tumor microenvironment [11,31,46,47]. There are several factors affecting CXCR4/ACKR3 signaling: ligand binding, receptor location and interactome [15], that might vary between cell types and physiological/pathological contexts.

The characterization of our selected BC cell line models revealed that distinct molecular subtypes of BC display very heterogeneous expression patterns of CXCR4, ACKR3, EGFR family receptors and GRKs. This heterogeneity is expected, given the already well described distinct molecular signatures of each cell model, but stresses the importance of studying CXCR4/ACKR3 signaling in its native context. Moreover, flow cytometry analysis indicates heterogeneous patterns of CXCR4/ACKR3 distribution, with a predominant localization of these receptors away from the plasma membrane and the presence of cell subpopulations with different proportions of these chemokine receptors at the cell surface, which suggests differential ability to bind and respond to CXCL12. Discrepancies regarding CXCR4/ACKR3 levels and distribution in BC cell lines are found in the literature [48,49,50,51]. These differences may arise from different sources of cells used, differing detection tools or changes in the cell culture, and maintaining protocols and thus in the cell environment that may alter receptor expression. The fact that in all BC lines a high proportion of CXCR4 and ACKR3 receptors was found internalized could be a result of enhanced steady-state receptor internalization. This intracellular pool might represent agonist-induced endocytosed receptors as a result of autocrine production and secretion of CXCL12 (triggered by CXCL12 secreted by tumor cells or arising from the TME) [11], a reservoir of naïve receptors, altered receptor trafficking processes or the existence of intracellular pools with a physiological function as suggested in normal settings of mammary gland development [52]. The specific localization of such intracellular receptors and whether they can elicit signaling responses remains to be investigated. In addition, co-expression of both CXCR4 and ACKR3 in the same cell is an open and relevant question since they could alter each other responses. In this regard, ACKR3 has the potential to alter CXCR4 responses by scavenging of CXCL12 (for which it displays a higher affinity than CXCR4), diverting the signal towards G protein-independent cascades (i.e., β-arrestin pathway) and/or preventing CXCR4 desensitization. While some reports suggest that both receptors co-localize [48], others have indicated that CXCR4 and ACKR3 are mostly expressed on separated populations of BC cells [49]. Understanding their distribution among subpopulations can unravel possible functional interactions at receptor level and clarify how uneven receptor localization modifies CXCR4 and ACKR3 transduction pathways. In mouse interneurons, ACKR3-mediated chemokine scavenging favors cell migration by impeding desensitization and downregulation of nearby CXCR4, thus potentiating CXCR4-dependent cascades [53]. In breast cancer, some reports suggest that both receptors are found in the same cell [49], whereas others report that CXCR4 and ACKR3 are found in distinct cell subsets, with uptake of CXCL12 by ACKR3+ BC cells enhancing proliferation and metastatic potential of CXCR4+ cells [50]. ACKR3 and CXCR4 in the same cell could also influence each other by heterodimerization (which remains to be confirmed in endogenous systems) [15], by competing for shared signaling proteins or via other crosstalk mechanisms. A recent study [33] has reported a highly heterogeneous response of cell subsets to CXCL12 (from strong to undetectable), which could reflect the heterogeneity we observed regarding receptor distribution. Overall, the emerging scenario suggests that the relative expression levels of CXCR4 and ACKR3 and whether they are expressed in the same or distinct subpopulations of cancer or stromal cells, may have an impact on CXCL12-mediated responses, highlighting a relevant question for future research.

Multiple signaling cascades are described as being stimulated by the CXCL12/CXCR4/ACKR3 axis [11,31,46,47]. We investigated the activation of the key ERK1/2 tumor progression cascade in response to exogenous CXCL12 in the different BC subtype models. Our data indicate that CXCL12-mediated ERK activation, although transiently more noticeable in MDA-MB-361 cells, was generally low in all cell lines. Interestingly, other key targets downstream CXCR4/ACKR3 such as PI3K/AKT, Src, p38 or S6, were not activated over steady-state levels in response to CXCL12. Such low response was not due to “saturation” of the pathway, since these cascades were robustly activated upon EGF stimulation, proving these pathways are still on dynamic range of stimulation. Interestingly, pharmacological inhibition of CXCR4 or (to a lesser extent) ACKR3 diminished AKT and c-Src activation, suggesting that these receptors basally signal to some pathways in the absence of exogenous CXCL12.

These results are consistent with the frequently reported modest effects on signaling readouts of extra CXCL12 in isolated and starved cancer cell types versus clearer effects upon CXCR4/ACKR3 receptor silencing or pharmacological inhibition ([11] and refs. therein). Such low response, and the variability observed among different BC cell lines, may arise from several factors. For instance, CXCL12 response patterns might be affected by cell-type-specific crosstalk with GF signaling networks and by the intrinsic oncogenic landscape of BC cell subtypes. Interestingly, our results show that CXCR4/ACKR3 display a positive crosstalk with EGFR, and more strongly with HER3/HER2, in MDA-MB-361 cells (luminal B subtype model, ER+, Her2-overexpressing, PI3KCA mutated), whereas concurrent activation of the CXCL12/CXCR4/ACKR3 and EGFR cascades attenuates ERK pathway activation pattern in MCF7 (luminal A model, ER+, PI3KCA mutated) or MDA-MB-231 (triple negative model, B-Raf and K-Ras mutated) cells. A recent study in BC cells, using a single cell approach with fluorescent receptor and Akt and ERK pathways activation reporters, demonstrated a highly heterogeneous response of cell subsets to CXCL12 (from strong to undetectable) and that changing tumor cell environmental inputs alters CXCR4 responsiveness [33]. The CXCR4 signaling status appears to be affected by driver mutations present in specific cells and then fine-tuned by recent conditioning of cells by growth factors. The heterogeneity of responsiveness to CXCL12 and the dependency on GF signaling described in this work is in line with the variability of responses observed in our cell lines and highlights the need to consider CXCR4/ACKR3 as part of complex signaling networks to fully understand how they can promote tumorigenesis.

Overall, these data suggested that EGFR family members and CXCR4/ACKR3 share signaling pathways, and that their crosstalk can lead to different outputs depending on different levels of expression of the ERbB receptors (higher in MD-MBA-361 cells), surface distribution of CXCR4 and ACKR3 and oncogenic landscape. Additionally, integration of multiple signaling inputs to the same pathway, rather than linear parallel pathways, also suggests the existence of common effectors downstream these receptor families. Our data put forward GRK2 as a relevant hub in EGFR family/CXCR4-ACKR3 crosstalk in BC cells. We observed that pharmacological inhibition of GRK2 in MD-MBA-361 cells did not alter the modest CXCL12-mediated ERK activation, but markedly reduced the robust ERK stimulation triggered by EGF or Heregulin alone or the fostered activation of this pathway detected upon simultaneous presence of CXCL12. A recent report in an overexpressing system demonstrated that even though GRK2 inhibition with cmpd101 decreased β-arrestin-2 recruitment to CXCR4 and ACKR3 it did not affect ERK1/2 activation by CXCL12-CXCR4 [54], suggesting that GKR2 was not required for CXCL12-dependent ERK activation, which is in line with our data. Of note, we find that extra levels of GRK2 protein tend to down-modulate ERK1/2 stimulation by CXCL12 alone, suggest a catalytic-independent desensitization mechanism, while leading to enhanced ERK1/2 stimulation by EGF alone, or upon co-stimulation with EGF and CXCL12. It is tempting to suggest that EGFR activation could relieve the scaffold-mediated inhibition exerted by GRK2 on CXCR4/ACKR3 signaling, by sequestering the protein away these receptors and/or via tyrosine phosphorylation mechanisms, overall switching GRK2 towards its ERK1/2 stimulation mode reported in other BC cell lines [25], leading to a stronger activation of this pathway. Whether this novel positive role in GPCR/RTK crosstalk is a general feature or depends on BC cell subtype and TME-derived factors requires further investigation though our data show a different behavior in MCF7 and MDA-MB-231 cells, strongly support the latter notion.

Our data indicate intricate crosstalk mechanisms between CXCR4/ACKR3 and EGFR-HER2 pathways in signaling to the ERK cascade in the MDA-MB-361 model (see scheme in Figure 7). Of note, AMD3100 or VUN400 nanobody-mediated CXCR4 inhibition (and, to a lesser extent, ACKR3 inhibition with the VUN700 nanobody) decreased EGF or HRG-mediated ERK activation suggesting that EGFR and HER3/HER2 receptors could signal to ERK1/2 partially through active CXCR4/ACKR3, even in the absence of extrinsic CXCL12, and that some components recruited by active chemokine receptors are essential for their crosstalk. However, we cannot rule out that autocrine production of CXCL12 might trigger CXCR4 activation and play a role in ERK1/2 stimulation by EGF or HRG alone. The effects of PTX are also consistent with the notion that EGF-RTKs signal partially through CXCR4 in a Gi-protein-dependent manner. Hijacking of the GPCR signaling machinery by GF-RTKs has been reported in other contexts [55,56]. Conversely, the effects of the EGFR tyrosine kinase inhibitor AG1478 slightly decreasing CXCL12-mediated ERK activation, indicates that a component of CXCR4/ACKR3 signaling depends on EGF-RTK.

The fact that fostered ERK phosphorylation by co-stimulation of CXCL12 and EGF or HRG is strongly attenuated upon c-Src inhibition also reveals an important role for this kinase in crosstalk, in agreement with c-Src being engaged downstream of either CXCR4/ACKR3 or EGFR-HER2/3 receptors stimulation [11,16]. EGF or HRG have been reported to promote CXCR4 serine and/or tyrosine phosphorylation in glioblastoma or BC cells leading to CXCL12-independent receptor activation of downstream cascades [16,57,58,59]. Conversely, the CXCL12/CXCR4/ACKR3 axis has been shown to enhance EGFR signaling indirectly by the release of EGFR ligands [60,61], or by triggering EGFR tyrosine phosphorylation and transactivation via Gi/Src-mediated pathways [62,63]. Since we find that GRK2 plays a relevant role in ErbB-CXCR4/ACKR3 crosstalk mechanism along with c-Src, it is tempting to suggest that these proteins are functionally related in this process. In support of this notion, both EGFR or GPCR/β-arrestin/Src cascades are known to converge in phosphorylating GRK2 on tyrosine residues [23,64], and GRK2 and EGFR can modulate CXCR4-mediated Gi activation in HEK-293 cells by mechanisms involving tyrosine phosphorylation [43].

In sum, our findings strongly suggest a relevant cooperation between CXCR4/ACKR3 and growth factor receptors in tumor progression, likely facilitated by the simultaneous increased abundance/functionality of these proteins in many tumor types. Since it is likely that the specific molecular landscape of BC cell subtypes (relative expression levels and activation status of ErbB family members, CXCR4/ACKR3 expression patterns and internalization extent, oncogenic mutations, interactors, spacio-temporal activation determinants, availability of downstream effectors) would be key in this regard [11,33], it would be important to further investigate these issues in endogenous cell model systems, to help design future therapeutic strategies targeting these receptor families. Understanding the mechanisms of communication between GPCRs and RTKs in tumor cells and other cells in the dynamic tumor microenvironment and its role in cancer growth, metastasis, and immune evasion is crucial to understand how to effectively target these receptors or downstream effectors in pathological processes.

## 4. Materials and Methods

### 4.1. Cell Culture and Cellular Treatments

All tumoral breast cells were obtained from the American Type Culture Collection (Manassas, VA, USA). MDA-MB-361 and MCF7 cells were maintained in DMEM supplemented with 10% (*v*/*v*) fetal bovine serum (FBS), glutamine 2 mM and penicillin/streptomycin (0.01%/0.063%) at 37 °C in a humidified 5% CO_2_ atmosphere. MDA-MB-231 cells were first cultured and amplified in L-15 medium supplemented with glutamine 2 mM and penicillin/streptomycin (0.01%/0.063%) and containing 15% (*v*/*v*) FBS at 37% in an CO_2_ null atmosphere and then adapted and maintained in DMEM medium supplemented with 10% (*v*/*v*) FBS, glutamine 2 mM and penicillin/streptomycin (0.01%/0.063%) at 37°C in a humidified 5% CO_2_ atmosphere. MDA-MB-361 cells were nucleofected with the nucleofection AmaxaTM Cell Line NucleofectorTM Kit V (Lonza, Basel, Switzerland) following manufacturer’s instruction. Briefly, 2 × 10^6^ cells were resuspended in 100 µL mixture of kit solution and 2 µg of pcDNA3-GRK2-WT or empty vector and electroporated using the program P-020. For signaling experiments cells at 80% confluence were serum-starved overnight (ON) (in the presence of 1% FBS). For cellular stimulation with ligands, cells were serum-starved overnight and challenged with Heregulin (20 ng/mL), EGF (100 ng/mL) or CXCL12 (10 nM) for the indicated times. To determine the influence of cellular components on the signaling dynamics of these ligands, cells were pretreated prior to stimulation with the inhibitors detailed in Figure legends and in Appendix A.

### 4.2. Immunoblotting

Proteins from cellular lysates were separated by SDS-PAGE electrophoresis on a 6–15% resolving gels, depending on experimental setup. Precision Plus Protein Standards Dual Color de BIO-RAD, was used as molecular weight maker. After electrophoresis, the proteins were transferred to nitrocellulose membranes (0.45 mm, BioRad, Hercules, CA, USA) using a wet blotting apparatus (Bio-Rad). The efficiency of transference was always evaluated by staining the membrane with Ponceau staining solution after blocking for 1 h in 5% BSA-TBS at room temperature. Afterwards, membranes were incubated with the primary antibodies diluted in 3% BSA-TBS, overnight at 4 °C (antibody details can be found in Appendix A). Next, the blots were washed three times for 10 min in TBS-Tween 0.1% followed by incubation with the secondary antibodies 1 h at room temperature. After extensive washing with TBS-Tween, membrane-bound secondary antibodies were detected using ECL (Enhanced ChemiLuminiscence, Amersham Pharmacia Biotech, Little Chalfont, UK) and Agfa films. Bands were quantified by laser densitometry with a Biorad GS-900 scanner and using de Bio-Rad provided Image Lab 5.2. Levels of phosphorylation-activated proteins were normalized either by total protein and equal protein loading was confirmed by comparing DJ1 levels as control.

### 4.3. Flow Cytometry

Flow cytometry was used to investigate cell distribution of CXCR4 and ACKR3. Cells were detached and fixed in 2%PFA in PBS for 20 min on ice. For conditions where intracellular receptor levels were analyzed, cells were permeabilized with 0.2% saponin in PBS-2%FBS. Afterwards, cells were incubated with primary antibody (5 × 10^5^ cells/condition; antibody dilution specified in methods section, prepared in PBS-2%FBS), for 45 min on ice. Cells were analyzed by flow cytometry in a FACS Canto II, and data obtained were processed with the FlowJo software (BD Life Sciences, Franklin Lakes, NJ, US).

### 4.4. mRNA Analysis and Quantification

Cultured cells were resuspended directly in TRIzol reagent (Invitrogen, Waltham, MA, USA) and mRNA was extracted using RNeasy Mini Kit (Qiagen, Hilden, Germany) following the instructions provided by the supplier. mRNA was quantified in a Nanodrop One spectrophotometer (Thermo Fisher Scientific, Waltham, MA, USA), the 260/280 and 260/230 ratio values were analyzed and discarded if they were far from a ratio of 2, which corresponds to pure RNA relative to protein contamination or to co-purified contaminants respectively. mRNA samples were further used for qRT-PCR analysis using the primers detailed in Appendix A. The qPCR experimental development and data analysis was provided by the Genomics and NGS Core Facility at the Centro de Biología Molecular Severo Ochoa (CBMSO, CSIC-UAM) which is part of the CEI UAM+CSIC, Madrid, Spain—http://www.cbm.uam.es/genomica (accessed 3 October 2022).

### 4.5. Quantification of CXCL12 Levels by ELISA

CXCL12 levels were determined in cell-conditioned media of BC cells plated on p100-dishes and incubated in DMEM 1% FCS for 48 h. A human CXCL12 Quantikine ELISA Kit (R&D Systems, Minneapolis, MN, USA) was used according to the manufacturer’s instructions. Protein levels of CXCL12 were calculated relative to the standard curve of the standard samples and were normalized to the amount of the total protein in cell lysates of seeded cells quantified by the Lowry method.

### 4.6. Statistical Analysis

Data are represented as mean ± SD of at least three independent experiments unless stated otherwise. In Figure legends, *n* represents the number of independent experiments. Statistical significance was determined by a 2-tailed Student’s *t*-test, by a one-way ANOVA followed by Dunnett’s multiple comparison test. All statistical analyses were completed with the GraphPad Prism software (GraphPad Software, La Jolla, CA, USA).

## Figures and Tables

**Figure 1 ijms-23-11887-f001:**
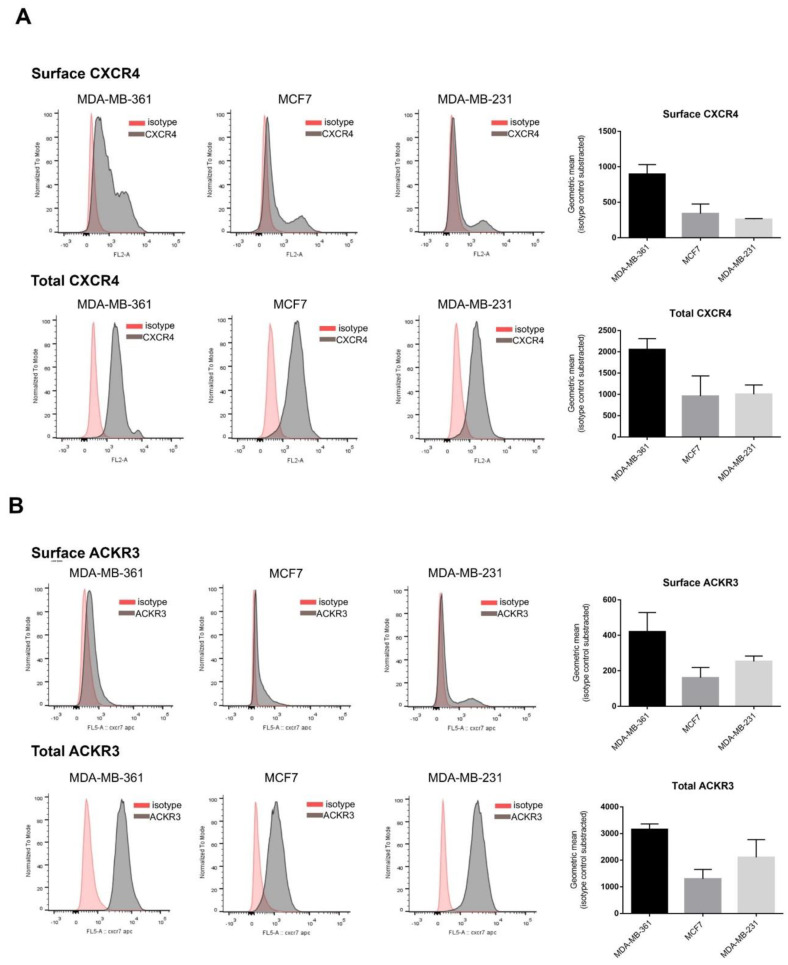
Model breast cancer (BC) cell lines display heterogeneous expression profiles of CXCR4/ACKR3 chemokine receptors, with a sizable intracellular pool and different proportions of cell surface receptors. Analysis of CXCR4 (**A**) or ACKR3 (**B**) expression and subcellular distribution was performed by flow cytometry analysis as detailed in the Methods for detection of surface receptors (upper plots) and total cellular receptors (surface + cytoplasmic) (lower plots) in MDA-MB-361 (ER+ and Her2+), MCF7 (ER+) and MDA-MB-231 (triple negative) cells. Data (geometric mean fluorescence corrected by the substraction of isotype control) are represented as mean of GMF ± SEM of 2–3 independent experiments.

**Figure 2 ijms-23-11887-f002:**
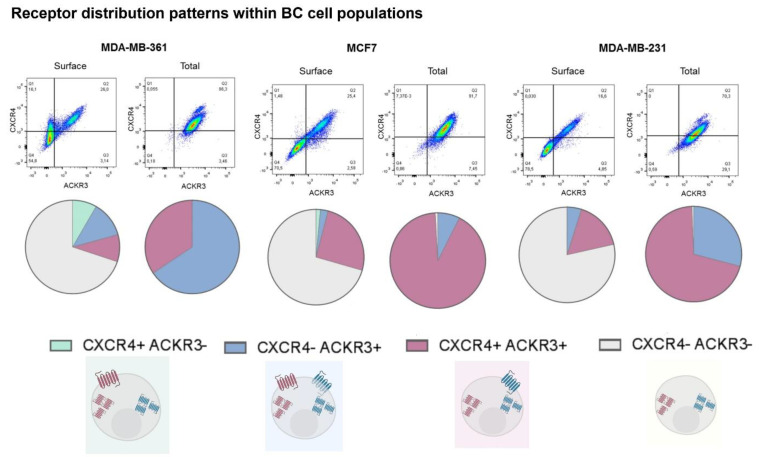
Model breast cancer cell lines display heterogeneous patterns of CXCR4/ACKR3 distribution and different cell subpopulations can be defined according to the concurrence or not of both receptors. Pie charts show percentage distribution of MDA-MB-361 (ER+ and Her2+), MCF7 (ER+) and MDA-MB-231 (triple negative) cells for the different combinations of CXCR4 and/or ACKR3 presence at the cellular surface or in the total cell for each cell line in 3 independent experiments.

**Figure 3 ijms-23-11887-f003:**
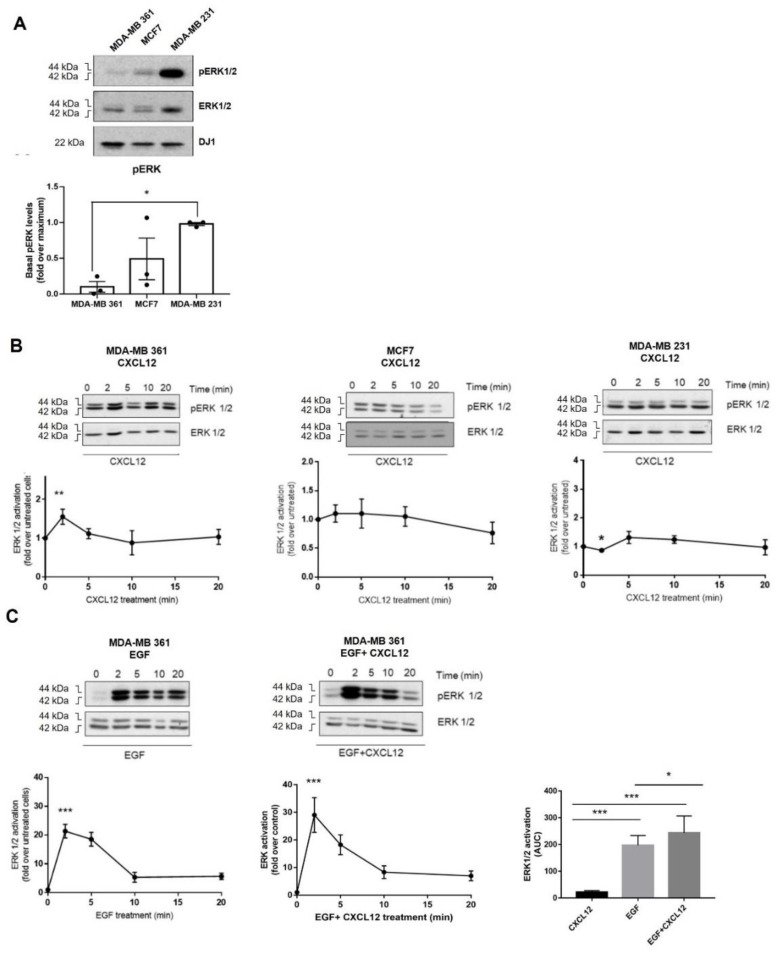
(**A**) Basal ERK1/2 activation levels in distinct BC cell lines. (**B**) ERK pathway is poorly stimulated in MCF7 (ER+), MDA-MB361 (ER+ and HER2+) and MDA-MB231 (triple-negative) cells in response to exogenous CXCL12. Serum-starved cells were stimulated with 10 nM CXCL12 for the indicated times and ERK1/2 activation assessed as detailed in Methods. Activated phospho-ERK1/2 was normalized by total ERK levels and referenced as fold change over untreated cells. Representative blots are shown. * *p* < 0.05, or *** *p* < 0.001, comparing to non-stimulated cells. (**C**) Simultaneous activation of the CXCL12/CXCR4/ACKR3 and EGFR pathways fosters the ERK cascade stimulation pattern in MDA-MB-361 cells (ER+ and HER2+). Serum-starved cells were stimulated with 10 nM CXCL12, 100 ng/mL EGF or the combination of both ligands for the indicated times. ERK1/2 activation was assessed and plotted as above along with the area under the curve (AUC). Representative blots are shown. Data are mean ± SEM of 9–13 independent experiments. ** *p* < 0.01, or *** *p* < 0.001, comparing to non-stimulated cells.

**Figure 4 ijms-23-11887-f004:**
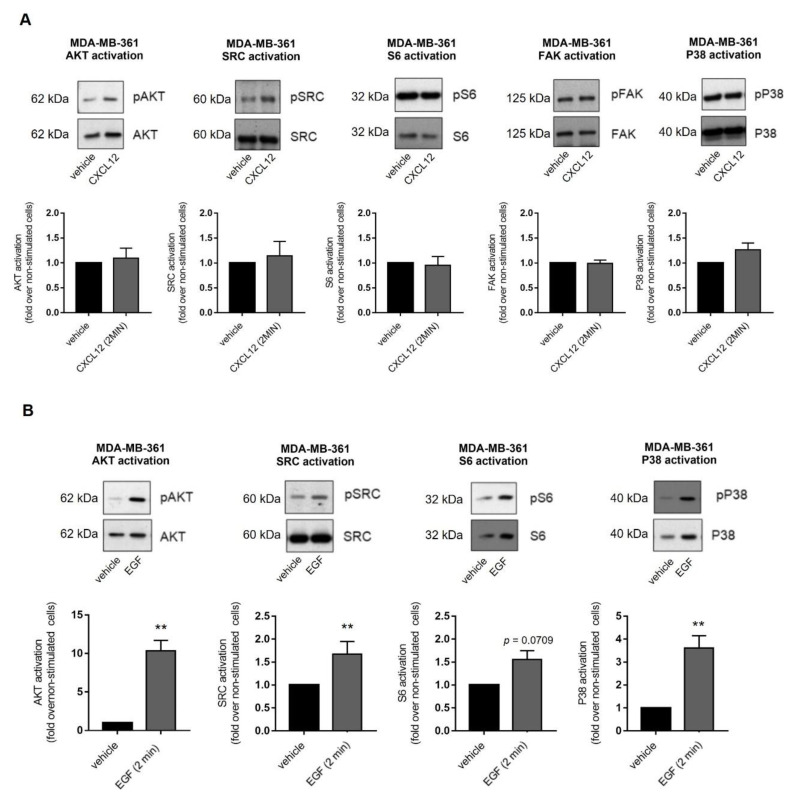
(**A**) Exogenous CXCL12 fails to further activate other cancer-related signaling pathways in MDA-MB-361 cells (ER+ and HER2+). (**B**) EGF strongly triggers key oncogenic signaling pathways in MDA-MB-361 cells. Cells were serum-starved and stimulated with 10 nM CXCL12 or 100 ng/mL EGF for 2 min. Activation of ERK1/2, AKT, SRC, S6 and p38 was measured in cell lysates by western blot with phospho-specific antibodies directed against the active proteins and data normalized by levels of total proteins. Representative blots are shown. Data are mean ± SEM of 2–10 independent experiments. ** *p* < 0.01, comparing to non-stimulated cells.

**Figure 5 ijms-23-11887-f005:**
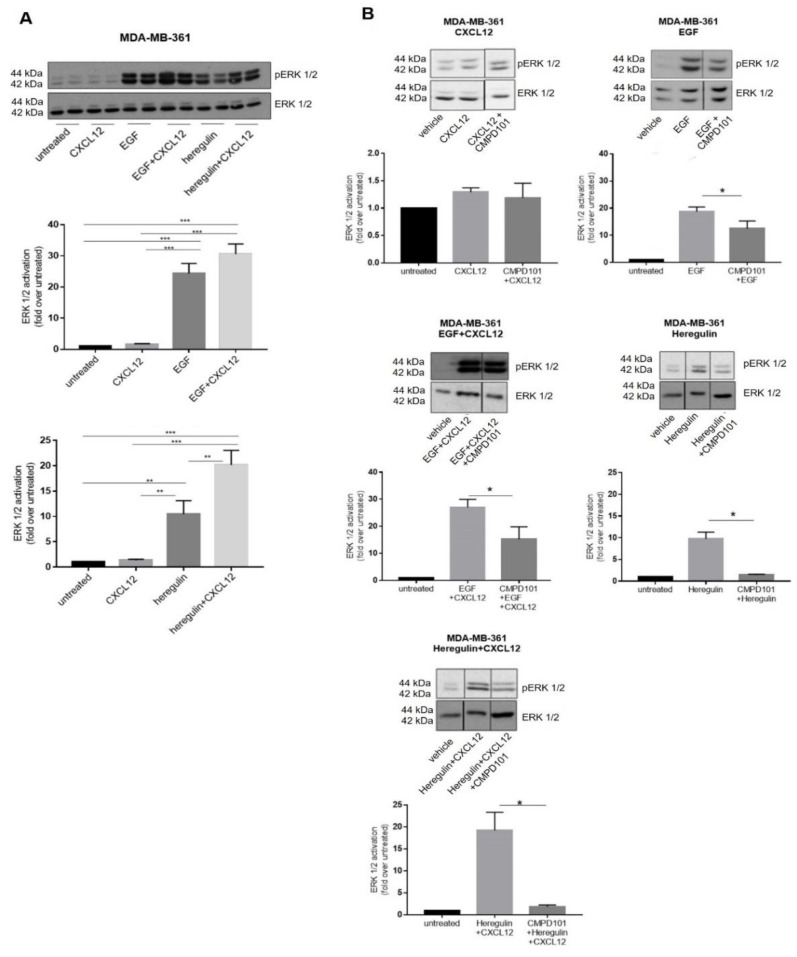
(**A**) Co-stimulation of CXCR4/ACKR3 with either EGFR or HER2/3 ligands amplifies ERK1/2 activation in MDA-MB-361 cells (ER+ and HER2+). Cells were serum-starved and stimulated with 10 nM CXCL12, 100 ng/mL EGF, 20 ng/mL Heregulin or the indicated combination of ligands for 2 min. ERK1/2 activation was assessed in cell lysates as in previous figures. Representative blots are shown. Data are mean ± SEM of 4–11 independent experiments. ** *p* < 0.01, *** *p* < 0.001 for the indicated comparisons. (**B**) Pharmacological inhibition of GRK2 markedly attenuates ERK activation promoted by EGF or Heregulin alone or in the simultaneous presence of CXCL12. Serum-starved MDA-MB-361 cells were treated for 2 h with 30µM CMPD101. Afterwards, cells were stimulated with 10 nM CXCL12, 100 ng/mL EGF, 20 ng/mL Heregulin or the indicated combinations for 2 min. ERK1/2 activation was assessed in cell lysates as in previous figures. Representative blots are shown. Blot lanes run on the same gel but non-contiguously are indicated by a black vertical line. Data are mean ± SEM of 4–9 independent experiments. * *p* < 0.05 for the indicated comparisons.

**Figure 6 ijms-23-11887-f006:**
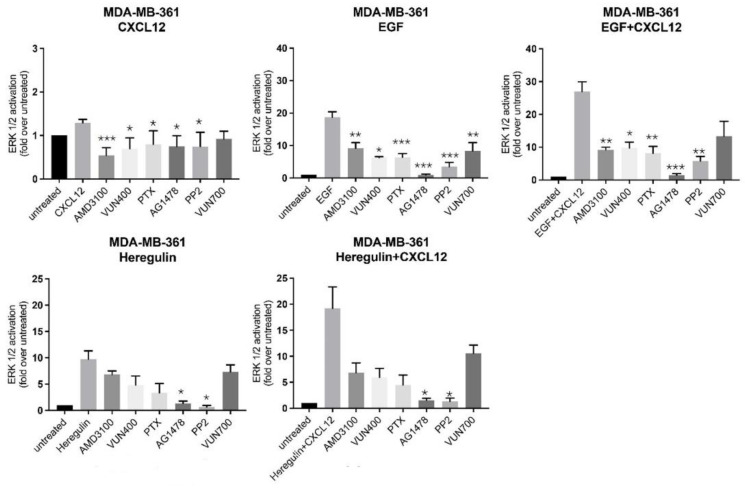
Complex crosstalk between CXCL12/CXCR4/ACKR3 and EGFR-HER2 pathways suggests shared mechanisms in signaling to the ERK cascade. MDA-MB-361 cells (ER+ and HER2+) were serum-starved and treated for 2 h with the indicated inhibitors for either CXCR4 (small-molecule compound AMD3100, nanobody V400), ACKR3 (nanobody VUN700), EGFR tyrosine kinase activity (AG1478), Gi protein (PTX) or Src kinase (PP2), as detailed in the Materials and Methods. Afterwards, cells were stimulated with 10 nM CXCL12, 100 ng/mL EGF, 20 ng/mL Heregulin or the indicated combinations for 2 min. ERK1/2 activation was assessed in cell lysates as above. Data are mean ± SEM of 2 independent experiments for VUN400 and 3–9 independent experiments for the remaining conditions. * *p* < 0.05, ** *p* < 0.01 or *** *p* < 0.001, comparing to stimulated cells without inhibitor treatment.

**Figure 7 ijms-23-11887-f007:**
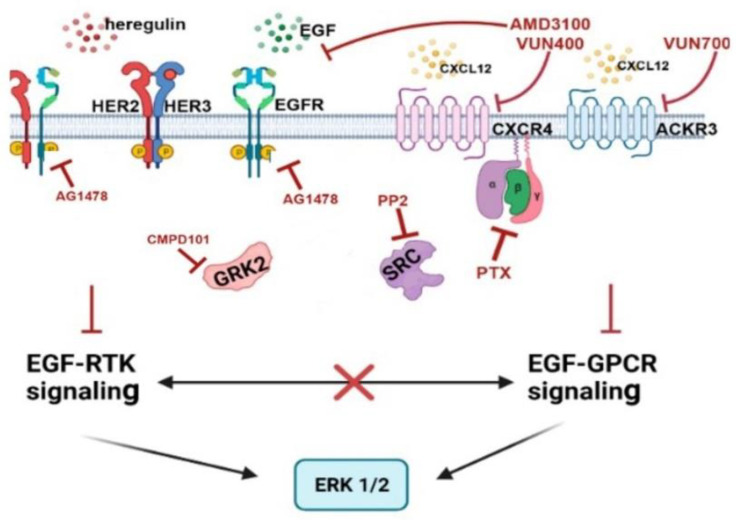
Schematic representation of the crosstalk mechanisms involved (see Discussion for details).

## Data Availability

Data generated or analysed during this study are included in this published article and its Appendix A.

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
