# Peer review of "Crosstalk between CXCR4/ACKR3 and EGFR Signaling in Breast Cancer Cells"

_ijms, 2022, doi:10.3390/ijms231911887_

Round 1

Reviewer 1 Report

Neves et al. investigated the interplay between the chemokine CXCL12/CXCR4/ACKR3 and EGF receptor family signaling cascades in three breast cancer subtypes. Their major finding is that CXCR4/ACKR3 and EGFR receptor families share signaling mechanisms activating ERK1/2 in the luminal B subtype but not in the luminal A or triple-negative subtype breast cancers. The experiments seem to be well designed and carefully performed. Plenty of mechanistic experiments have been done. The topic is interesting, however, the presentation does not always help the reader. Generally, the text is hard to follow and needs to be simplified and shortened. To improve the quality of the presentation and readability, I have the following suggestions:

1. The authors mention the classification of breast cancer subtypes basd on the expression of key genes in the Introduction. However, the difference between luminal A (ER+) and luminal B (ER+, HER2 overexpression) subtypes is mentioned only in the discussion. Please introduce these differences in the Introduction part and clarify why luminal A, luminal B, and triple-negative cell lines were used in this study (epidemiology, prevalence, or resistance reasons?)

2. It would be really helpful if the clinically used receptor subtypes (ER+ after MCF7; ER+ and HER2+ after MDA-MB-361; triple-negative after MDA-MB-231) would be written after the name of the breast cancer cell line on the graphs and Fig. legends.

3. To help the general understanding, the Fig. 4B panel should be shown separately as Fig. 1 and summarize the receptors and mechanisms (including MAPK/ERKs) depicted here in the second part of the Introduction.

4. The Results part should not contain long introductions and explanations of the results (e.g., page 3, lines 99-112, 117-126, page 5, lines 164-175, etc.). Please shorten and put these parts into the Discussion.

5. Generally, there are too many panels on the figures, and therefore, the numbers and letters on graphs are hardly readable. There is no limit for the number of Figures in this journal, therefore, I suggest separating the big Figures as much as possible (e.g., Fig. 1 and Fig. 4) and writing about the results in shorter form. It will be easier to follow the results if the text is put near to the Figures.

6. Please remove the description of the detailed methods from the Figure legends and put them into the Methods section.

7. Please magnify the WB bands and put the representative WB images over the bar graphs. Please show also the hosuekeeping protein or total form of each protein (e.g., Fig. 3 and Fig. 4). Indicate the size of each protein on the Figures.

8. Figure 4E should be separated and show as a summary Figure at the end of the MS.

9. At the beginning of the discussion, the novelty of the study should be summarized in a few sentences. At the end of the discussion, a clinical relevance and perspectives section is needed to emphasize the relevance of the findings.

Author Response

ANSWERS TO THE COMMENTS OF THE REVIEWERS

REVIEWER 1

Comments and Suggestions for Authors

Neves et al. investigated the interplay between the chemokine CXCL12/CXCR4/ACKR3 and EGF receptor family signaling cascades in three breast cancer subtypes. Their major finding is that CXCR4/ACKR3 and EGFR receptor families share signaling mechanisms activating ERK1/2 in the luminal B subtype but not in the luminal A or triple-negative subtype breast cancers. The experiments seem to be well designed and carefully performed. Plenty of mechanistic experiments have been done. The topic is interesting, however, the presentation does not always help the reader. Generally, the text is hard to follow and needs to be simplified and shortened. To improve the quality of the presentation and readability, I have the following suggestions:

  1. The authors mention the classification of breast cancer subtypes basd on the expression of key genes in the Introduction. However, the difference between luminal A (ER+) and luminal B (ER+, HER2 overexpression) subtypes is mentioned only in the discussion. Please introduce these differences in the Introduction part and clarify why luminal A, luminal B, and triple-negative cell lines were used in this study (epidemiology, prevalence, or resistance reasons?)

As suggested by the reviewer, the key differences in molecular profiling among breast cancer subtypes (lines 50-55) and the rationale for choosing luminal A, luminal B and triple-negative cell models in our study (lines 100-102) are now mentioned in the Introduction section and at the beginning of the Results section (lines 110-112).

  1. It would be really helpful if the clinically used receptor subtypes (ER+ after MCF7; ER+ and HER2+ after MDA-MB-361; triple-negative after MDA-MB-231) would be written after the name of the breast cancer cell line on the graphs and Fig. legends.

In line with the suggestion of the reviewer, the clinical notation corresponding to the different breast cancer cell models used in this study has been incorporated into the Figure legends.

  1. To help the general understanding, the Fig. 4B panel should be shown separately as Fig. 1 and summarize the receptors and mechanisms (including MAPK/ERKs) depicted here in the second part of the Introduction.

To help with the general understanding and following the suggestion of point 5 of this reviewer, we have reorganized several Figures and panels, including former Fig. 4B. Please see our answer to point 5 below.

  1. The Results part should not contain long introductions and explanations of the results (e.g., page 3, lines 99-112, 117-126, page 5, lines 164-175, etc.). Please shorten and put these parts into the Discussion.

An effort has been made to streamline the Results section as suggested by the reviewer, and move key points to the Discussion section.

  1. Generally, there are too many panels on the figures, and therefore, the numbers and letters on graphs are hardly readable. There is no limit for the number of Figures in this journal, therefore, I suggest separating the big Figures as much as possible (e.g., Fig. 1 and Fig. 4) and writing about the results in shorter form. It will be easier to follow the results if the text is put near to the Figures.

We thank the reviewer for his/her suggestion, and Figures have been reorganized to make them clearer and more readable:

-Former Fig. 1C panel is now Figure 2.

-Former Figure 2 (now Figure 3) has been simplified by moving panel D to Supplementary Figure 2A.

-Former Figure 4 has been reorganized as follows: former panels A and C are now Figure 5; former panel D is now Figure 6, and former panel E is now Figure 7.

Please also note that, to accommodate the suggestions of other reviewer, an additional graph has been added to Fig. 3C, and additional data have been included as Supplementary Figure S1B (control of endogenous CXCL12 secretion by the different cell subtypes), and as Supplementary Figure S4 (effect of GRK2 over-expression on the response of MD-MB-361 cells to different combinations of stimuli)

  1. Please remove the description of the detailed methods from the Figure legends and put them into the Methods section.

As suggested by the reviewer, most of the experimental details have been transferred from the legends to the Methods section.

  1. Please magnify the WB bands and put the representative WB images over the bar graphs. Please show also the hosuekeeping protein or total form of each protein (e.g., Fig. 3 and Fig. 4). Indicate the size of each protein on the Figures.

As indicated by the reviewer, the representatives WB images are now shown over the bar graphs in new Figures 3, 4 and 5. The size of each protein (the phosphorylated and total forms of the analysed signalling kinases and housekeeping controls) is now indicated in the blots.

  1. Figure 4E should be separated and show as a summary Figure at the end of the MS.

As explained in point 5, in line with this suggestion panel E from former Figure 4 is now Figure 7, summarizing the main findings at the end of the manuscript.

  1. At the beginning of the discussion, the novelty of the study should be summarized in a few sentences. At the end of the discussion, a clinical relevance and perspectives section is needed to emphasize the relevance of the findings.

A brief novelty statement has been incorporated at the beginning of the Discussion section. The potential clinical relevance of the findings and future perspectives and lines of research were already stated at the end of the Discussion; however, a brief sentence has been incorporated to further emphasize this issue.

Reviewer 2 Report

In this manuscript, Neves M et. al. studied the role of CXCR2/ACKR3 (CXCR7) and EGFR signaling for their cross-talk interaction focusing majorly on ERK1/2 activation event in Breast Cancer cells. This signaling interaction may be important to understanding BC dynamics and heterogeneous responses, however, this manuscript has several critical issues (see comments below) that need to be clearly addressed and warrants major revision.  

Major commments.

1. Fig 1. line141-144. The data in fig 1 shows the variable expression of CXCR4 and ACKR3. It doesn't support the author's conclusion about ligand accessibility and population distribution. 

2. Fig 1A, B. Analyzing fold change of CXCR4 protein expression over isotype control Ab geo-MFI is confusing and does not lead to a meaningful comparison of protein expression on the different cell lines used. Instead, the authors need to show geo-MFI values of anti-CXCR4 staining on these cells.  Also, whether the authors have evaluated the dose-response curve of various BC cells treated with CXCL12? how 10 nM concentration is optimal?  

3. Fig S2 shows the transient activation of ERK in response to EGF alone or EGF+CXCL12. However, the authors conclude that ERK phosphorylation is attenuated, which is contrasting with what data in Fig S2 shows (at least in MCF7 cells).

4. Data from figure 2 and S2 does not support or suggest any direct link between the ERK activation pattern with variability in surface expression of CXCR4 or CXCR7. The author's conclusion of this data on lines 200-202 is truly speculative. 

5. Fig 2C. It looks like the signaling from EGF alone leads to the robust activation of ERK1/2. The ERK activation in response to the combination of EGF+CXCL12 looks similar to EGF alone. The authors need to statistically compare the activation of ERK with EGF versus EGF+CXCL12 to clearly show if the combination is superior in its ability to invoke ERK activation. Similar comparisons need to be performed for fig 2D.   

6. Why did some of these cell culture experiments perform more than 10 times? 

7. Fig 4C. As in fig 2, the attenuation of ERK activation in response to GRK2 inhibitor shows that the inhibitor can downmodulate the ERK activation in EGF+CXCL12 or CXCL12+Heregulin treated cells but does not have activity in CXCL12 alone treated cells. A similar level of downmodulation is seen for EGF or heregulin-alone treated cells. This data does not support the author's conclusion that GRK2 is needed for cross-talk between CXCR4/ACKR3 and EGF or HRG signals. This experiment in fig 4C does not directly address any issue with the signaling cross-talk but rather shows that GRK2 can inhibit ERK activation signal via EGF and HRG.  

8. What is the status of ERK activation upon inhibiting both CXCR4 and ACKR3? In other words, whether blocking one of these renders cells to use other available receptors for CXCL12? 

9. Does the BC cells used in the study secrete CXCL12 in the culture? Did the authors try to measure if any CXCL12 was secreted in the culture? 

10. Authors need to discuss the physiological significance of this data in the context of tumor growth, proliferation, metastasis, or it any immune evasion processes. How cooperation between these signaling events helps tumor cells to meet their needs in the dynamic tumor microenvironment?

Minor comments. 

1. Spell out all the abbreviations used in the text.

2. There are several typo and grammar errors throughout the text that needs attention. 

Author Response

ijms-1839726

ANSWERS TO THE COMMENTS OF THE REVIEWERS

REVIEWER 2

Comments and Suggestions for Author

 In this manuscript, Neves M et. al. studied the role of CXCR2/ACKR3 (CXCR7) and EGFR signaling for their cross-talk interaction focusing majorly on ERK1/2 activation event in Breast Cancer cells. This signaling interaction may be important to understanding BC dynamics and heterogeneous responses, however, this manuscript has several critical issues (see comments below) that need to be clearly addressed and warrants major revision.  

We appreciate the positive comments of the reviewer and his/her criticisms to help improve our manuscript. Detailed responses are given below.

As a general introductory remarks, please note that, following the suggestions of other reviewer, Figures have been reorganized to make them clearer and more readable:

-Former Fig. 1C panel is now Figure 2.

-Former Figure 2 (now Figure 3) has been simplified by moving panel D to Supplementary Figure 2A.

-Former Figure 4 has been reorganized as follows: former panels A and C are now Figure 5; former panel D is now Figure 6, and former panel E is now Figure 7.

Also, as requested by other reviewer, representative WB images are now shown over the bar graphs in new Figures 3, 4 and 5, and the size of each protein is indicated in the blots.

In addition, to accommodate your suggestions an additional graph has been added to Fig. 3C, and additional data have been included as Supplementary Figure S1B (control of endogenous CXCL12 secretion by the different cell subtypes), and as Supplementary Figure S4 (effect of GRK2 over-expression on the response of MD-MB-361 cells to different combinations of stimuli). Please see details below.

Major commments.

  1. Fig 1. line141-144. The data in fig 1 shows the variable expression of CXCR4 and ACKR3. It doesn't support the author's conclusion about ligand accessibility and population distribution. 

Text in former lines 141-144 has been modified to better explain this issue. The point we wanted to make is now summarized as follows (lines 152-155):

Overall, our results put forward a very variable presence of surface CXCR4 and ACKR3 across different BC cell lines and complex patterns of CXCR4/ACKR3 distribution and co-localization within cell lines, suggesting that the response of BC cells to CXCR4/ACKR3 ligands might be very heterogenous.

  1. Fig 1A, B. Analyzing fold change of CXCR4 protein expression over isotype control Ab geo-MFI is confusing and does not lead to a meaningful comparison of protein expression on the different cell lines used. Instead, the authors need to show geo-MFI values of anti-CXCR4 staining on these cells.  Also, whether the authors have evaluated the dose-response curve of various BC cells treated with CXCL12? how 10 nM concentration is optimal?  

Following the suggestion of the reviewer, geo-MFI values are now shown in the bar graphs of Figure 1A-B. Regardless of this comparison, the key qualitative point that all BC cell lines subtypes display a high proportion of internalized CXCR4 and ACKR3 receptors and that the MDA-MB-361 cell line expresses slightly more CXCR4 in the plasma membrane compared to the other subtypes.

Regarding the dose response to CXCL12, we performed preliminary experiments that suggested that 10 nM CXCL12 was a suitable concentration (see a representative example in the Figure below for the consideration of the reviewer). Highest chemokine concentrations have been correlated with increased receptor desensitization in other experimental settings.

Figure for the reviewer. Serum-starved MCF7 cells were stimulated with the indicated concentrations of  CXCL12 for 5 min and ERK1/2 activation assessed by western blot with specific antibodies as detailed in Methods.

  1. Fig S2 shows the transient activation of ERK in response to EGF alone or EGF+CXCL12. However, the authors conclude that ERK phosphorylation is attenuated, which is contrasting with what data in Fig S2 shows (at least in MCF7 cells).

We stated in our previous manuscript that “Conversely, in MCF7 and MDA-MB-231 cells, no potentiation of ERK1/2 phosphorylation, but rather an attenuation of the activation pattern of this pathway was observed when EGF and CXCL12 were combined”

Thus, we wanted to note that, in addition to changes observed in the extent of the peak of ERK stimulation, ERK1/2 activation was more transient and decayed faster in MCF7 cells, whereas a more general attenuation was present in MDA-MB-231 cells. We have modified the text in the Results section and in the Figure legends to better explain these data shown in Fig. S2B of the revised version. The Results text now reads:

“Conversely, in MCF7 and MDA-MB-231 cells, no potentiation of ERK1/2 phosphorylation was observed when EGF and CXCL12 were combined (Figure S2B), but rather a faster decay (MCF7) or attenuation (MDA-MB-231) of the activation pattern of this pathway.”

Moreover, in order to stress the changes taking place in the pattern of ERK1/2 stimulation, an analysis of the area under the curve (AUC) of these experiments has been added to Figures 3C and S2B when analysing the effects of CXCL12, EGF/heregulin or combined stimulations in MD-MB-361 cells.

  1. Data from figure 2 and S2 does not support or suggest any direct link between the ERK activation pattern with variability in surface expression of CXCR4 or CXCR7. The author's conclusion of this data on lines 200-202 is truly speculative.

Lines 200-202 of the former version of the manuscript did read:

“ …suggesting that signalling amplification effects between these receptors not only re-quire high levels of EGF but a cell-type specific repertoire of receptors and downstream transducers and regulators”

Therefore, we were not suggesting a direct link between the ERK activation pattern and surface expression of CXCR4/ACKR3. We raised the point that the positive cooperation between CXCL12 and EGF appears only to be observed in certain BC cell subtypes and is not a general phenomenon. When we refer to a “cell-type specific repertoire of receptors and downstream transducers and regulators”, we are not limiting to the expression of CXCR4 and ACKR3, but also considering the dynamic range of ERK activation, or the expression levels of members of the EGF receptor family. To make this point clearer we now state (lines 205-211):

“This suggests that signaling amplification effects between these receptors not only re-quire high levels of EGF but a cell-type specific repertoire of EGFR and HER2·receptors and downstream transducers and regulators. We thus decided to focus our research on CXCR4/ACKR3 signaling and crosstalk mechanisms in the MDA-MB-361 BC cell model, a subtype characterized by lower basal levels of active ERK and high levels of EGFR and HER2 heterodimers, and which displayed a positive crosstalk of ErbB-RTKs and CXCR4/ACKR3 to the ERK pathway.”

In addition, the Discussion on the potential impact of CXCR4/ACKR3 surface expression, cell co-localization and distribution heterogeneity on signalling responses has been also modified to better explain this issue (lines 395-402).

  1. Fig 2C. It looks like the signaling from EGF alone leads to the robust activation of ERK1/2. The ERK activation in response to the combination of EGF+CXCL12 looks similar to EGF alone. The authors need to statistically compare the activation of ERK with EGF versus EGF+CXCL12 to clearly show if the combination is superior in its ability to invoke ERK activation. Similar comparisons need to be performed for fig 2D.   

As suggested by the reviewer, we have analysed in more detail the changes in the pattern of ERK1/2 stimulation in response to CXCL12, EGF or Heregulin or CXCL12/EGF or CXCL12/Heregulin combinations by quantifying the area under the curve (AUC) of these experiments and statistically compare such experimental conditions. This approach indicates that simultaneous stimulation of MDA-MB-361 cells with EGF and CXCL12 promotes a higher ERK1/2 activating phosphorylation pattern, with a significant increase in the area under the curve (AUC) compared to EGF or CXCL12 alone (new graph in Figure 3C), with a similar trend was observed upon co-stimulation with heregulin (Figure S2A).

  1. Why did some of these cell culture experiments perform more than 10 times? 

The comparative nature of many of the experimental systems and the need for adequate statistical analysis in some conditions.

  1. Fig 4C. As in fig 2, the attenuation of ERK activation in response to GRK2 inhibitor shows that the inhibitor can downmodulate the ERK activation in EGF+CXCL12 or CXCL12+Heregulin treated cells but does not have activity in CXCL12 alone treated cells. A similar level of downmodulation is seen for EGF or heregulin-alone treated cells. This data does not support the author's conclusion that GRK2 is needed for cross-talk between CXCR4/ACKR3 and EGF or HRG signals. This experiment in fig 4C does not directly address any issue with the signaling cross-talk but rather shows that GRK2 can inhibit ERK activation signal via EGF and HRG.  

To further address the role of GRK2 in the crosstalk among CXCR4/ACKR3 and EGF receptor family, we have used an experimental approach additional to the use of the CMPD101catalytic GRK2 inhibitor, by means of heterologous expression of extra levels of wild-type GRK2 in MDA-MB-361 cells (data shown in new Supplementary Fig S4 and mentioned in the Results and Discussion sections).

Interestingly, extra levels of GRK2 protein tend to down-modulate ERK1/2 stimulation by CXCL12 alone (Figure S4A), consistent with its previously reported role in CXCR4/ACKR3 desensitization [27].  The fact that pharmacological inhibition of GRK2 has no apparent effect on CXCL12 signaling (Figure 5B) may suggest a catalytic-independent desensitization mechanism. GRK2 overexpression enhances ERK1/2 stimulation by EGF alone, and co-stimulation with EGF and CXCL12 in GRK2-overexpressing cells leads to a significantly stronger activation of ERK1/2 compared to each stimulus alone (Figure S4B-C-D). These results support an overall positive role for GRK2 in ERK1/2 activation when both CXCL12 and EGFR are co-stimulated. It is tempting to suggest that EGFR activation could relieve the scaffold-mediated inhibition exerted by GRK2 on CXCR4/ACKR3 signaling, by sequestering the protein away these receptors and/or via tyrosine phosphorylation mechanisms, overall switching GRK2 towards its ERK1/2 stimulation mode, leading to a stronger activation of this pathway.

  1. What is the status of ERK activation upon inhibiting both CXCR4 and ACKR3? In other words, whether blocking one of these renders cells to use other available receptors for CXCL12? 

As shown in Figure S3, inhibition of CXCR4 or ACKR3 does not have an effect on basal ERK activation. CXCR4 inhibitors decrease CXCL12-mediated ERK activation, whilst ACKR3 blocking has no  significant effect, suggesting that blocking this receptor does not facilitate CXCL12/ ERK signaling via the alternative CXCL12 receptor CXCR4 (Fig. 6)

  1. Does the BC cells used in the study secrete CXCL12 in the culture? Did the authors try to measure if any CXCL12 was secreted in the culture? 

Yes, we had addressed this issue, and the data have been incorporated in new Fig. S1B and mentioned in the text as follows (lines 122-126):

The quantification of CXCL12 in cell conditioned media (Figure S1B) indicates that all BC lines secrete different levels of this chemokine, although these do not overall correlate with the extent of internalized receptors, thereby suggesting that other signaling components in BC cancer cells might contribute to shape a steady-state pool of intra-cellular receptors.

  1. Authors need to discuss the physiological significance of this data in the context of tumor growth, proliferation, metastasis, or it any immune evasion processes. How cooperation between these signaling events helps tumor cells to meet their needs in the dynamic tumor microenvironment?

Following the suggestion of the reviewer, a brief sentence in this regard has been incorporated at the end of the Discussion section.

Minor comments. 

  1. Spell out all the abbreviations used in the text.

We have revised and spelled out the first time they appear in the text the non-common abbreviations as suggested.

  1. There are several typo and grammar errors throughout the text that needs attention. 

The text has been revised for typo and grammar errors.

Round 2

Reviewer 1 Report

The authors answered all of my questions and suggestions.

I recommend the acceptance of their modified MS for publication in IJMS.